# Infective Endocarditis in People Who Inject Drugs: Report from the Italian Registry of Infective Endocarditis

**DOI:** 10.3390/jcm11144082

**Published:** 2022-07-14

**Authors:** Enrico Cecchi, Silvia Corcione, Tommaso Lupia, Ilaria De Benedetto, Nour Shbaklo, Fabio Chirillo, Antonella Moreo, Mauro Rinaldi, Pompilio Faggiano, Moreno Cecconi, Olivia Bargiacchi, Alessandro Cialfi, Stefano Del Ponte, Angelo Squeri, Oscar Gaddi, Maria Gabriella Carmina, Alessandro Lazzaro, Giovannino Ciccone, Anna Castiglione, Francesco Giuseppe De Rosa

**Affiliations:** 1Department of Cardiology, Maria Vittoria Hospital, 10144 Turin, Italy; cecchi.enrico@tin.it; 2Department of Medical Sciences, Infectious Diseases, University of Turin, 10124 Turin, Italy; silvia.corcione@unito.it (S.C.); ilaria.debenedetto@edu.unito.it (I.D.B.); nour.shbaklo@edu.unito.it (N.S.); alessandro1lazzaro@gmail.com (A.L.); francescogiuseppe.derosa@unito.it (F.G.D.R.); 3Unit of Infectious Diseases, Cardinal Massaia Hospital, 14100 Asti, Italy; 4Department of Cardiology, Ca’ Foncello Hospital, 31100 Treviso, Italy; fabio.chirillo@aulss7.veneto.it; 5Department of Cardiac Surgery, Molinette Hospital, University of Torino, 10124 Torino, Italy; antonella.moreo@ospedaleniguarda.it; 6Department of Cardiology, Niguarda Ca’ Granda Hospital, 20124 Milan, Italy; mauro.rinaldi@unito.it; 7Department of Cardiology, Fondazione Poliambulanza, 25124 Brescia, Italy; deicas@med.unibs.it; 8Department of Cardiac Sciences and Surgery, Azienda Ospedaliera Universitaria, Ospedali Riuniti, 60126 Ancona, Italy; moreno.cecconi@sanita.marche.it; 9Infectious Diseases Section, Maggiore della Carità Hospital, 28100 Novara, Italy; olivia.bargiacchi@gmail.com; 10Department of Cardiology, Sacco Hospital, 20157 Milano, Italy; cialfi.alessandro@hsacco.it; 11Department of Cardiac Surgery, Mauriziano Hospital, 10128 Torino, Italy; sdelponte@mauriziano.it; 12UO Cardiologia, Maria Cecilia Hospital, GVM Care & Research, 48033 Cotignola, Italy; angelo.squeri@yahoo.it; 13Centro Cuore Salute, 42124 Reggio Emilia, Italy; gaddi.oscar@asmn.re.it; 14Department of Cardiology, United Hospitals of Palermo, 90146 Palermo, Italy; m.carmina@ospedaliriunitipalermo.it; 15Unit of Clinical Epidemiology, CPO, City of Sciences and Health, 10126 Turin, Italy; gciccone@cittadellasalute.to.it (G.C.); anna.castiglione@cpo.it (A.C.)

**Keywords:** infective endocarditis, PWID, HIV, endovascular infections, IVDU

## Abstract

Intravenous drug use is a predisposing condition for infective endocarditis (IE). We report the clinical features of IE, taken from the Italian Registry of IE, in people who inject drugs (PWIDs). The registry prospectively collected epidemiological, clinical, in-hospital, and follow-up data on patients with IE from 17 Italian centers. A total of 677 patients were enrolled, and 61 (9%) were intravenous drug users (IDUs). Most PWIDs were male (78.6%), and aged between 41 and 50 years old (50%). The most frequent comorbidities were HIV (34.4%) and chronic liver disease (32%). Predisposing factors for IE were present in 6.5% of the patients, and 10% had minor valvular abnormalities. IE had occurred previously in 16.4% of the patients, and 50% of them had undergone heart surgery. Overall mortality was 9.8% in IDUs and 20% in patients with recurrent IE. IE in PWIDs mostly affected the native valves (90%). The echocardiographic diagnosis of IE was based on the detection of vegetation in 91.82% of cases. *Staphylococcus aureus* was the main microorganism isolated (70%) from blood cultures. Thirty patients (49%) underwent heart surgery: thirteen had aortic valves, eleven had mitral valves, and six had tricuspid valve interventions. IE in PWIDs was relatively common, and patients with native valve right-sided IE had a better prognosis, with a low rate of surgical interventions.

## 1. Introduction

Intravenous drug use is a predisposing condition for infective endocarditis (IE) and represents a minor diagnostic criterion for diagnosis [1]. In the last decade, the incidence of IE in people who inject drugs (PWIDs) has sometimes declined because of a general switch from intravenous to other routes of drug intake (i.e., intranasal, smoking, or oral) as well as damage reduction policies implemented in the 1990s for HIV prevention [2]. However, there are reports, especially in the United States, of increasing hospitalization for IE in PWIDs among the younger white female demographic [3].

Currently, IE in the general population remains a relatively rare disease, with a high in-hospital mortality rate, ranging from 15 to 26% depending on the case series and an even greater morbidity rate, due mainly to heart failure or valvular vegetations with high embolic potentials [2,3]. Prognostic improvements are difficult and should be focused on early diagnosis, multidisciplinary management, and optimization of the timing and techniques of cardiac surgery [4,5]. Historically, PWIDs with right-sided endocarditis have consistently been considered to have better prognoses, potentially with shorter antibiotic therapy courses, outpatient treatment, and ad hoc strategies of cardiosurgery. Unfortunately, the literature shows that more than one-third of PWIDs may present with left-sided IE and a dismal prognosis [2,3,5,6,7].

Data recently published by the Italian Registry of Infective Endocarditis (RIEI) indicated that, despite continuous epidemiological changes, IE in PWIDs still represents 9% of IE enrolled in the RIEI [2]. We report here the epidemiological and clinical features of IE in PWIDs from the RIEI.

## 2. Materials and Methods

RIEI prospectively collected epidemiological, clinical, in-hospital, and follow-up data on patients with IE established according to the modified Duke criteria [1] from 17 Italian centers coordinated by the Cardiology Unit of Maria Vittoria Hospital in Turin.

RIEI has prospectively enrolled consecutive cases of infective endocarditis in every participating center, analyzing diagnostic and therapeutic data from a real world practice perspective. No experimental interventions were proposed, and treatments were entirely decided by treating physicians. Enrollment required the following features: (1) high clinical suspicion of acute and/or active infective endocarditis (definite or highly possible infective endocarditis according to Duke criteria), (2) consecutive recruitment and prospective evaluation at presentation, (3) sufficient data availability to fulfill enrolment and follow-up records, and (4) enrollment of consecutive cases and strict monitoring of suspected cases in the laboratory of echocardiography as well as in the microbiology laboratory for identification of microorganisms. For the purposes of the present study, only cases with definite IE according to modified Duke criteria and complete echocardiographic data have been included.

Complete methods and data acquisition and storage methods, together with definitions and procedures, have already been published [2].

## 3. Results

Of the 677 patients enrolled in the RIEI, 61 were intravenous drug users (IDUs) (9%). The main clinical characteristics and comorbidities, compared with the remaining patients in the RIEI registry, are reported in Table 1. PWIDs were predominantly male (78.6%), middle-aged (50% of PWIDs were between 41 and 50 years old), and with chronic liver disease (32%) or chronic heart disease (28%). Only 6.5% (4) of patients had predisposing factors for IE, and 10% had minor valvular abnormalities (Table 1).

Previous cases of IE had occurred in 16.4% (10) of patients, and 50% of them underwent heart surgery. Overall mortality was 9.8% in IDUs, but it rose to 20% in patients with recurrent IE.

IE in PWIDs mostly affected the native valves (55; 90%), especially the left-side valves (29; 53%). Prosthetic valve IE was reported in 6.5% of patients (four; two aortic and two tricuspid) and in two cases with endocavitary devices. Twenty-one PWIDs (34.42%) were HIV-positive, and 33% of them had a CD4 level <200/mm^3^.

The most prominent sign was a fever of >38 °C, which was present in 100% of IE cases. Clinical vascular embolic events were found in 18% of patients upon physical examination, while imaging studies disclosed pulmonary infarction in 21% of patients. An acute clinical course led to immediate hospital admission in 40% of patients, and within 7 days in 31% of patients. In both instances, the rate of hospital admission after medical consultation was faster than for the other patients in the RIEI. Echocardiographic diagnoses of IE were based on the detection of vegetation in 91.82% of patients and on a myocardial abscess in 13%.

Blood cultures were positive in 82% of cases, and *Staphylococcus aureus* was the main microorganism isolated (70%), with a methicillin resistance rate of 12.5% (Table 2).

The second most frequent microorganism was *Streptococcus viridans* (10%), usually occurring in patients with poor oral hygiene (78%). Infections were mostly community-acquired (53; 87%), and only six cases (9.8%) were nosocomial.

Thirty patients (49%) underwent heart surgery: thirteen with aortic valve replacements (three associated with mitral valve surgery), eleven with mitral valve interventions (four repairs) of which two were associated with tricuspid valve replacement, and six with further tricuspid valve interventions (four replacements). The indications for tricuspid valve surgery were pulmonary embolism with a large residual vegetation in two cases, bacteremia or persistent infection in two cases, and severe tricuspid regurgitation and large vegetation in two cases. All patients undergoing tricuspid valve surgery had severe valvular insufficiency. Many of the 22 cases undergoing left-sided valve surgery had multiple indications; severe or moderate-severe valvular insufficiencies and vegetations > 10 mm occurred in 17 cases (77%). In addition, 13 patients (59%) had heart failure, 10 (45%) had previous embolic events, 5 (22%) presented with myocardial abscesses, and 3 (13%) had persistent bacteremia. Pulmonary infarction was found in 26% of PWID cases, occurring more frequently than in IE cases in the general population (Table 3).

Twenty-six patients had right-side IE (twenty-four with native valves and two with prosthetic valves). In this group, all patients had high fever, but none had neurological symptoms. Three had systemic embolic events, seven presented with heart failure (three right, two left, and two mixed), ten had septic pulmonary infarction, ten presented with severe tricuspid valve insufficiencies, and eight (30%) underwent tricuspid valve surgery (three plastic repairs). None had pulmonary oedema. There was only one death in this group (3.8%) observed in a HIV-positive patient with low CD4+ lymphocyte count.

Ten out of the twenty-one HIV-positive patients presented with heart failure (one right, five left, and four mixed), which was severe in five cases. In-hospital complications occurred in 14 patients, mostly with stroke, followed by systemic embolic events and septic pulmonary infarction. Eleven of the twenty-one patients underwent cardiosurgery (52%), mostly for aortic or mitral valve replacements (five and five, respectively). The in-hospital mortality rate for this group was 19%.

## 4. Discussion

Even in the absence of heart disease, IDU is considered a predisposing condition for IE in developed countries, but the epidemiology and clinical manifestations of IE in PWIDs are continuously evolving [2,3]. PWIDs account for 9% of the patients enrolled in RIEI, a higher percentage than that recently found by Ortiz-Bautista et al. and Alkhawam et al., with 5% and 4.3%, respectively, but slightly lower than the data reported in a prospective survey conducted in Piedmont, Italy, in 2001 [5,6,8].

In contrast to some reports, but in agreement with other studies, the overall prognosis of IE in PWIDs is still variable, despite the high frequency of heart surgery (49%). It is slightly lower than in the general IE population, with a mortality rate of 9.8%, rising to 20% in relapsing events [6,9]. As reported in previous studies, our findings show that the mortality rate is very low (3.8%) in patients with isolated right-sided IE, even though almost 30% of patients in this group underwent heart surgery [10]. As in other reports, HIV-seropositive patients in the RIEI cohort had a less favorable prognosis, with a mortality rate of 19%, despite a high rate of referral for heart surgery (52%) in Italian centers for patients with left heart involvement [6,11]. Moreover, as expected, relapsed IE is associated with higher in-hospital mortality.

Our data highlight some new features of IE in PWIDs, such as the increasing median age of patients; they are older than previously reported (41 to 50 years old), probably because they are being referred to care later in life when comorbidities may be present. Indeed, 6.1% of patients already had IE, 10% had prosthetic heart valves, and two patients had endocavitary devices. Notably, more than 30% of cases are HIV-seropositive, with poor responses probably due to a lower compliance with treatment. In our study, HIV-positive PWIDs with IE were up to 50 years old and had a prolonged history of intravenous drug use and more advanced heart disease than in the past [12]. These facts account for the frequent complications and higher mortality rates, as opposed to those of non-IDU IE patients of more advanced age, who more commonly present with left heart involvement. Relapsing IE is associated with previous surgeries, left-side IE, and continuous intravenous drug abuse. It has a high mortality rate. These data highlight the need for evidence-based protocols aimed at managing drug abuse to reduce relapses and improve outcomes in these patients. This brief report has several limitations. First, RIEI is an observational study performed in both referral and non-referral centers for IE, with the prevalent involvement of Cardiological and Cardiosurgical wards as opposed to Infectious Disease wards. Potential selection bias is probably present in these settings, as 45% of patients are referred to study centers from other hospitals. Furthermore, due to the brief nature of this report, some data on treatment regimens and a complete dataset with stratifications for mortality are lacking.

## 5. Conclusions

In conclusion, IE in PWIDs was relatively common in our registry, and patients with native valve right-sided IE had a better prognosis, with a low rate of surgical interventions. The mortality rate was high in patients who experienced a relapse due to prolonged use of injected opioids, left heart involvement, or a positive HIV status. Our preliminary data add to the growing body of evidence demonstrating increasing concerns about substance misuse and injection-mediated risk and suggest the need for public health interventions and clinical policies to manage drug abusers. Further analyses are needed to complete the evaluation of these preliminary findings.

## Figures and Tables

**Table 1 jcm-11-04082-t001:** Patient characteristics and comorbidities.

	PWIDs *n* (%)	Non-PWIDs *n* (%)	Total *n* (%)
**Sex**			
Male	48 (78.69)	444 (27.92)	492 (27.33)
Female	13 (21.31)	172 (72.08)	185 (72.67)
**Age Class (years)**			
<40	28 (45.9)	52 (8.44)	80 (11.82)
41–50	31 (50.82)	61 (9.9)	92 (13.59)
51–60	2 (3.28)	93 (15.1)	95 (14.03)
61–70	0	157 (25.49)	157 (23.19)
71–80	0	193 (31.33)	193 (28.51)
>80	0	60 (9.74)	60 (8.86)
**Ongoing IDU**	61 (100)	0	61 (9.01)
**History of IDU**	61 (100)	22 (3.75)	83 (12.26)
**Comorbidities**			
Cancer	1 (1.64)	65 (10.55)	66 (9.75)
COPD	1 (1.64)	64 (10.39)	65 (9.6)
History of acute myocardial infarction	0	47 (7.63)	47 (6.94)
Ischemic heart diseases	1 (1.64)	94 (15.26)	95 (14.03)
Hypertension	1 (1.64)	256 (41.56)	257 (37.96)
Renal failure	4 (6.56)	81 (13.15)	85 (12.56)
Alcoholism	1 (1.64)	21 (3.41)	22 (3.25)
Peripheral arterial disease	1 (1.64)	46 (7.47)	47 (6.94)
Chronic liver disease	20 (32.79)	44 (7.14)	64 (9.45)
Peptic ulcer	1 (1.64)	17 (2.76)	18 (2.66)
Neurologic vascular diseases	0	35(5.68)	35(5.17)
Stroke	0	29(4.71)	29 (4.28)
HIV-positive	21(34,42)	19 (3.06)	40 (5.8)
AIDS	9 (14.75)	7(1.1)	16 (2.3%)
**Procedures**			
Dental	1 (1.64)	31 (5.03)	32 (4.73)
Intravenous catheter	6 (9.84)	152 (24.68)	158 (23.34)
Other invasive procedures	4 (6.56)	135 (21.75)	149 (22.01)

Abbreviations: IDU: intravenous drug use; PWID: people who inject drugs; COPD: chronic obstructive pulmonary disease; AIDS: acquired immunodeficiency syndrome; HIV: human immunodeficiency virus.

**Table 2 jcm-11-04082-t002:** Blood culture results and microorganisms isolated in positive blood cultures.

	PWIDs *n* (%)	Non-PWIDs *n* (%)	Total *n* (%)
**Positive blood culture**	50 (81.97)	442 (71.75)	492 (72.67)
*Staphylococcus aureus*	35 (70)	98 (22.17)	133 (27.03)
*Staphylococcus epidermidis*	4 (8)	57 (12.9)	61 (12.4)
Coagulase-negative *Staphylococci*	1 (2)	43 (9.73)	44 (8.94)
*Streptococcus bovis*	0	57 (12.9)	57 (11.59)
*Streptococcus viridans*	5 (10)	68 (15.38)	73 (14.84)
*Staphylococcus lugdunensis*	0	3 (0.68)	3 (0.61)
*Enterococcus faecalis*	3 (6)	67 (15.16)	70 (14.23)
HACEK group	0	6 (1.36)	6 (1.22)
*Enterococcus faecium*	0	5 (1.13)	5 (1.02)
*Escherichia coli*	0	11 (2.49)	11 (2.24)
*Klebsiella* spp.	0	4 (0.9)	4 (0.81)
*Pseudomonas* spp.	3 (6)	1 (0.23)	4 (0.81)
**Negative blood culture**	8 (13.11)	161 (26.14)	169 (24.96)

Abbreviations**:** PWIDs: people who inject drugs; HACEK group: *Haemophilus* species, *Aggregatibacter* species, *Cardiobacterium hominis*, *Eikenella corrodens,* and *Kingella* species.

**Table 3 jcm-11-04082-t003:** IE site and complications.

	PWIDs *n* (%)	Non-PWIDs *n* (%)	Total *n* (%)
**IE site**			
Native valve	55 (11.17)	437 (88.82)	492 (100)
Prosthetic valve	6 (3.24)	179 (96.75)	185 (100)
**Common complications associated with IE**			
Pulmonary infarction	16 (26.23)	32 (5.19)	48 (7.09)
Pulmonary edema	9 (14.75)	65 (10.55)	74 (10.93)
Intracranial hemorrhage	4 (6.56)	28 (4.55)	32 (4.73)
Mycotic aneurysm	4 (6.56)	21 (3.41)	25 (3.69)
Pulmonary infarction	16 (26.23)	32 (5.19)	48 (7.09)
Pulmonary edema	9 (14.75)	65 (10.55)	74 (10.93)
Intracranial hemorrhage	4 (6.56)	28 (4.55)	32 (4.73)
Mycotic aneurysm	4 (6.56)	21 (3.41)	25 (3.69)

IE: infective endocarditis; PWID: People who inject drugs.

## Data Availability

The data presented in this study are available upon request from the corresponding author.

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
