# Peer review of "Infective Endocarditis in People Who Inject Drugs: Report from the Italian Registry of Infective Endocarditis"

_jcm, 2022, doi:10.3390/jcm11144082_

Round 1
Reviewer 1 Report
what was the chronic liver disease in each group.
what was the chronic lung disease (COPD)?
High incidence of cancer in Non PWIDS why?
Details of antibiotic regeimes used for treatment and for how long
Author Response
- What was the chronic liver disease in each group
Dear reviewer, thank you for your comment. We defined chronic liver disease as a progressive deterioration of liver functions for more than six months, and we take into account different etiologies such as toxins, alcohol abuse for a prolonged time, infection, autoimmune diseases, and genetic and metabolic disorders. This is a Brief Report from the Italian Registry and this is an interesting point that will be developed in the second part of the analysis, despite our early analysis defining an incidence of about 49% of chronic liver disease due to hepatitis B virus or C virus infection (or both), which is the primary aetiology in this cohort. We will s
- What was the chronic lung disease (COPD)?
Dear reviewer, thank you for your comment that improves our manuscript. We have changed “Chronic Lung Diseases” into “COPD-Chronic Obstructive Pulmonary Disease”, which constitutes 100% of the causes of lung disease in this cohort. We have made the changes in the text.
- High incidence of cancer in Non PWIDS why?
Dear reviewer, thank you for this comment. Cancer is an immunosuppressive disorder that could increase the risk of infection, and this risk could be increased during chemotherapies and targeted therapies. Moreover, cancer patients undergo a substantial amount of invasive procedures during diagnosis and treatment, which puts them at risk of developing IE. In patients without a history of PWID, in our opinion, cancer remains one of the main risk factors for developing IE.
- Details of antibiotic regeimes used for treatment and for how long
Dear reviewer, thank you for your comment. This is a Brief Report from the Italian Registry and this is an interesting point that will be developed in the second part of the analysis and could be part of a full article, despite the that our early analysis defined a slight prevalence of glycopeptide and lipoglycopeptides (31%) in the regimens for the antibiotic treatment of IE.
Reviewer 2 Report
This is a well-designed study from the Italian Registry of Infective Endocarditis that prospectively collected data on patients with IE from 17 Italian centers. The manuscript presents data on 61 patients with intravenous drug use, about 9% of all recruited patients in the Registry. Half of them needed surgery, while patients with right-heart IE had a better prognosis.
Some specific comments can be found below:
1. Line 44: PWID is not a common abbreviation, thus, even though it is okay to keep that in the manuscript, consider adding IVDU as a keyword to make the manuscript more discoverable
2. Line 75: IDU. Please write in full the terms when first abbreviated in the text
3. Table 1: Define PWID, HIV and AIDS at the footnotes
4. Table 2: Define PWID at the footnotes. And delete IDUs (I can’t see that in this table)
5. Table 3: Define PWID at the footnotes. And delete IDUs (I can’t see that in this table)
6. A table with data on site of infection (affected valve), mortality and maybe treatment duration could allow the reader to more easily find this information visually. This table could be constructed like Table 1, and could even be merged with Table 3
7. Limitations of the study should be mentioned at the end of the discussion section, right before the conclusion section. For example, selection bias could be a limitation
8. Methods may have been described in detail in the previous manuscript that is shown in the reference, but still could be described in brief herein. For example, it is important to know how IE was defined (were only definite included? Were possible included as well?). This is of particular importance as more than 10% of IE cases in IDU are shown to be culture negative. How was the diagnosis performed in those cases?
9. The authors could provide a statistical analysis among the PWID and non-PWID in the tables, to show the statistical significant differences among the groups
Author Response
Reviewer comments
Reviewer 2#
This is a well-designed study from the Italian Registry of Infective Endocarditis that prospectively collected data on patients with IE from 17 Italian centers. The manuscript presents data on 61 patients with intravenous drug use, about 9% of all recruited patients in the Registry. Half of them needed surgery, while patients with right-heart IE had a better prognosis.
Some specific comments can be found below:
- Line 44: PWID is not a common abbreviation, thus, even though it is okay to keep that in the manuscript, consider adding IVDU as a keyword to make the manuscript more discoverable
Dear reviewer, thank you for this comment. We added IVDU in the keywords according to your suggestion
- Line 75: IDU. Please write in full the terms when first abbreviated in the text
Dear reviewer, thank you for this comment. We added the term intravenous drug user in the text according to your suggestion
- Table 1: Define PWID, HIV and AIDS at the footnotes
Dear reviewer, thank you for this comment. We added these abbreviations in the footnotes of Table 1 according to your suggestions
4)Table 2: Define PWID at the footnotes. And delete IDUs (I can’t see that in this table)
Dear reviewer, thank you for this comment. We added these abbreviations in the footnotes of Table 1 according to your suggestions
- A table with data on site of infection (affected valve), mortality and maybe treatment duration could allow the reader to more easily find this information visually. This table could be constructed like Table 1, and could even be merged with Table 3
Dear reviewer, thank you for these comments. We have modified Table 3 according to your suggestions, despite that complete data on mortality in each group and data on treatment are not completely analyzed and will be one of the issues of a future full article from this registry. We have added these limitations in the discussion section.
- Limitations of the study should be mentioned at the end of the discussion section, right before the conclusion section. For example, selection bias could be a limitation
Dear reviewer, thank you for this comment. We added limitations in the discussion section according to your suggestions
- Methods may have been described in detail in the previous manuscript that is shown in the reference, but still could be described in brief herein. For example, it is important to know how IE was defined (were only definite included? Were possible included as well?). This is of particular importance as more than 10% of IE cases in IDU are shown to be culture negative. How was the diagnosis performed in those cases?
Dear reviewer, thank you for this comment. We have expanded the methods section to better define the selection criteria also for culture-negative IE. You can find corrections, highlighted in the text according to your suggestions
- The authors could provide a statistical analysis among the PWID and non-PWID in the tables, to show the statistical significant differences among the groups
Dear reviewer, thank you for these comments. Despite that complete statistical analysis is not available due to the brief report nature of the manuscript. The complete analysis will be published and will be one of the issues of a future full article from this registry. We have added these limitations in the discussion section.
Round 2
Reviewer 2 Report
The manuscript has been improved during the revision process.